# Cheese and Butter as a Source of Health-Promoting Fatty Acids in the Human Diet

**DOI:** 10.3390/ani12233424

**Published:** 2022-12-05

**Authors:** Beata Paszczyk

**Affiliations:** Department of Commodity Science and Food Analysis, The Faculty of Food Science, University of Warmia and Mazury in Olsztyn, 10-719 Olsztyn, Poland; paszczyk@uwm.edu.pl; Tel.: +48-89-523-36-81

**Keywords:** cheeses, butter, fatty acids, odd- and branched-chain fatty acids, conjugated linoleic acid (CLA), *trans* isomers, lipid quality indices

## Abstract

**Simple Summary:**

Cheese and butter are dairy products that are very popular among consumers. Technological progress has led to a multitude of cheese types available on the market, varying in texture and flavor. Cheeses are an important source of a wide variety of biologically active substances, among which specific fatty acids are of utmost importance. Butter is the oldest animal fat known to man and occupies an important place among the fat products on the market. Butter is a product with high nutritional value and health-promoting properties, as well as great flavor and aroma values. The quality of milk fat is determined, primarily, by the composition of its fatty acids, the properties of which depend on the length of the carbon chain and the presence of unsaturated bonds. The fatty acid composition of cheese and butter varies according to milk origin (e.g., species and breed), rearing conditions (e.g., feeding), and, in the case of cheese, it also depends on cheese-making technology. This work presents the fatty acid composition, including the contents of CLA and *trans* C18:1 and C18:2 isomers, in fat extracted from cheeses and butters available on the Polish market and the comparison of their lipid quality indices.

**Abstract:**

The assessment of fatty acid composition, including the content of conjugated linoleic acid *cis*9*trans*11 C18:2 (CLA) and *trans* C18:1 and C18:2 isomers in fat extracted from selected high-fat dairy products commonly available to consumers in retail sale on the Polish market, and a comparison of their indicators as to the quality of lipids was the aim of the study. The experimental materials were hard cheeses, white-mold cheeses, blue-veined cheeses, and butters. The conducted study demonstrated that various contents of groups of fatty acids and the values of lipid quality indices were found in the tested products. Butters turned out to be richer sources of short-chain, branched-chain, and odd-chain fatty acids. The fat extracted from butters and white-mold cheeses had a significantly higher (*p* < 0.05) content of *n*-3 fatty acids. Lower values of the *n*-6/*n*-3 ratio were determined in the fat extracted from butters and white-mold cheeses. The highest values of the thrombogenicity index (TI) were found in fat extracted from hard cheeses. Significantly lower values (*p* < 0.05) of the atherogenicity index (AI) and values of the H/H ratio were found in fat from mold cheeses. Fat from butters and white-mold cheeses had a significantly higher (*p* < 0.05) content of CLA and total content of *trans* C18:1.

## 1. Introduction

Milk and dairy products are an important dietary source of essential nutrients, such as protein, vitamins, minerals, and fat, including various fatty acids (FAs) [1,2]. Milk fat consists of more than 400 different fatty acids and is the most complex fat of all fats found in the human diet [3,4]. The quantitative composition of fatty acids in cow’s milk changes under the influence of multiple factors, including the feeding system of the animals, breed of the cows, lactation period, individual characteristics and health status of cows, and others [5,6]. Out of these factors, the most significant effect is ascribed to the feeding system [7,8,9,10,11,12]. In the human diet, milk and dairy products are important sources of saturated fatty acids (SFA), including the C12:0, C14:0, and C16:0 acids. These products also contain other fatty acids that have a beneficial effect on our health, including butyric acid (C4:0), branched fatty acids (BCFA), odd fatty acids (OCFA), oleic acid (*cis*9 C18:1), and the conjugated linoleic fatty acid *cis*9*trans*11 C18:2 (CLA) [13]. Milk fat is also the richest dietary source of natural *trans* fatty acid isomers, mainly vaccenic acid (*trans*11 C18:1, VA), which has opposite beneficial properties compared to artificial *trans* fatty acids in partially hydrogenated oils [14,15]. This acid, constituting over 50% of all *trans* isomers of C18:1 acid in milk fat, has been shown to exhibit anticarcinogenic and antiatherosclerosis effects [16].

Cheese and butter are dairy products that are very popular among consumers [17]. Cheese is defined as the fresh or matured product obtained from the coagulation of milk. There are many types of cheese on the market today. These cheeses differ, among other things, in texture and flavor. The cheese types can be classified according to the milk used (cow, sheep, goat, buffalo), their manufacture (rennet, sour milk cheese), consistency (extra-hard, hard, semi-hard, semi-soft, soft, fresh cheese), fat content (double cream, cream, full fat, three-quarters fat, half fat, quarter fat cheese), fermentation type (lactic acid, lactic and propionic acid, butyric acid), and surface (hard, soft, with smear, molds) [13]. Additionally, cheeses differ in flavor as well as some bioactive components formed during different stages of ripening when the main ingredients, i.e., lactose, protein, and fat, are broken down by fermentation, proteolysis, and lipolysis [18,19,20]. Cheeses are an important source of a wide range of biologically active substances, among which certain fatty acids are of particular importance. Mold cheeses belong to a specific group of ripened rennet cheeses which, when ripened in suitable conditions, develop exquisite flavors, aroma, and texture resulting from the presence of selected microbiological bacteria and production methods [21,22,23,24]. They fall into the category of rennet cheeses made from milk and rennet. Depending on the method of production adopted, they are divided into white-mold cheeses, such as Camembert and Brie, and blue-green-veined cheeses, such as Roquefort. The first are produced with white mold cultures of *Penicillium camemberti* or *Penicillium candidum* and feature a soft texture, creamy color, no holes, melting in the mouth, and a mild aroma, as well as a mushroom, slightly spicy, bitter-sour taste. In turn, strains of the species *Penicillium roqueforti*, *Penicillium gorgonzola*, or *Penicillium glaucum* are used to make the blue-veined cheeses characterized by mold growth throughout the bulk of the cheese, which has a semi-soft consistency. After a few weeks of ripening, these cheeses acquire a mild, slightly sour mushroom aftertaste. As they ripen, their taste and aroma become sharper, and their consistency turns more crumbly. In Poland, mold cheeses are less popular than hard cheeses. They can be eaten as elements of sandwiches, they can be used as ingredients in salads and casseroles, or they can also be consumed as a snack [25]. Mold cheeses have a high energy value, and due to their high-fat content (from 24% to 32%), they also are a great source of calcium [26]. Butter is the oldest animal fat known to man and occupies an important place among the fat products on the market. It is a high-fat product manufactured exclusively from cow’s milk as a result of the so-called churning of specially prepared sour or sweet cream. Both cream and cream intended for the production of butter should contain at least 25–35% fat [27]. Butter is a product with high nutritional value and health-promoting properties, as well as great flavor and aroma values. It is a high-fat product containing at least 80% but not more than 90% fat. Butter is a readily purchased product and its greatest advantages, according to consumers, include its taste, its health-promoting properties, and the fact that it is a natural product [28]. The quality of milk fat is determined, primarily, by the composition of its fatty acids, the properties of which depend on the length of the carbon chain and the presence of unsaturated bonds. The fatty acid composition of cheese and butter varies according to milk origin (e.g., species and breed), rearing conditions (e.g., feeding), and, in the case of cheese, it also depends on cheese-making technology (e.g., coagulation process, the addition of salt, ripening period) [19]. The fatty acids of the cheese fat are necessary for the appropriate sensory properties of the cheese and for the development of its flavor during ripening [24].

The composition of fatty acids in dairy products, including hard cheeses made from the milk of various ruminants, has been the subject of many studies [29,30,31,32,33,34,35,36,37,38], whereas other reports have focused on the assessment of the fatty acid profile in blue cheeses [36,38,39,40] and butter [36,41,42,43,44]. A rich assortment of butter and cheese available on the Polish market is largely attributed to the high intake and popularity of those products among Polish consumers. Taking into account the fact that the composition of these products may change as a result of changes in the quality of the raw material or various parameters used in technological processes, the constant monitoring of their quality is of great importance from a nutritional point of view. Therefore given that the fatty acid profile of milk and dairy products is an important factor affecting their nutritional value, the aim of this study was to determine the composition of fatty acids, with special emphasis on the concentrations of conjugated linoleic acid *cis*9*trans*11 C18:2 (CLA), *trans* C18:1 and C18:2 isomers, and the lipid quality indices in selected cheeses (hard and mold) and butters commonly available to consumers in retail sale on the Polish market.

## 2. Materials and Methods

### 2.1. Samples

The research materials were 40 samples of different dairy products with high fat content, including 10 samples of hard cheeses with contents of fat ranging from 26% to 28% (Gouda, Edamski, Podlaski, Królewski, Salami, Włoszczowski, Edam, Sokół, Złoty Mazur), 10 samples of white-mold cheeses containing from 24% to 32% of fat (Brie Mlekovita, Brie Turek, Brie Valbon, Brie Hochland, Brie President, Camembert Mlekovita, Camembert Turek, Camembert Valbon, Camembert President, Camembert Hochland), 10 samples of blue-veined cheeses with the fat content of 29% to 32% (NaTurek GrandBlue, Lazur Blękitny, Lazur Srebrny, Valbon Blue, Rokpol, Carrefour Classic, Mlekovita La Polle Polish Bleu, Turek Blue, Bakoma Regnum Blue, Pilos Blue), and 10 samples of butter with 82% fat (Ekstra Polmlek, Ekstra OSM w Kole, Ekstra łowickie, Ekstra ze Sztrzałkowa, Mazurskie, Polskie, Ekstra Piątnica, Ekstra Mazurski Smak, Ekstra Pilos, Wiejskie). All of the analyzed products were made from cow’s milk, produced by various manufacturers in Poland, and were available on the Polish market from June to July. Each sample was analyzed in duplicate. 

### 2.2. Methods 

#### 2.2.1. Lipid Extraction

Folch’s method was used to extract the fat from the analyzed cheeses [45]. Fat from butter was separated by melting at 40 °C, decantation, and filtering through anhydrous sodium sulfate.

#### 2.2.2. Preparation of Fatty Acid Methyl Esters

Fatty acids were converted into the corresponding fatty acid methyl esters (FAME) according to the IDF standard method (ISO 15884:2002) [46].

#### 2.2.3. Analysis of Fatty Acid Composition by GC Method

The methyl esters obtained in the process were then analyzed by gas chromatography (GC). Chromatographic separation was performed using a Hewlett-Packard 6890 gas chromatograph (Műnster, Germany) with a flame-ionization detector (FID) and a capillary column with a length of 100 m and an internal diameter of 0.25 mm (Chrompack, Middelburg, The Netherlands). The liquid phase was CP Sil 88 and the film thickness was 0.20 μm. The conditions of separation were as follows: carrier gas: helium, 1.5 mL/min flow rate; column temperature: 60 °C, 5 °C/min increase to 180 °C; detector temperature: 250 °C; injector temperature: 225 °C. The sample injection volume was 0.4 μL (split: 50:1). The identification of fatty acid methyl esters was carried out by comparing their retention times with the retention time of the fatty acid methyl esters of the reference milk fat (BCR Reference Materials) with the CRM 164 symbol (Aldrich, Taufkirchen, Germany) and for the C18:1 and C18:2 positional *trans* isomers, the methyl ester standards of these isomers (Sigma-Aldrich, Germany and Supelco, Bellefonte, PA, USA) were used. Amounts of fatty acids, CLA contents, and *trans* C18:1 and C18:2 isomers were calculated in mg/g of fat with respect to the introduced standard (methyl ester of C21:0 acid).

#### 2.2.4. The Lipid Quality Indices

The lipid quality indices Index of Atherogenicity (AI) and Index of Thrombogenicity (TI) were calculated according to the fatty acid composition using the following formulae [47,48]:AI = (C12:0 + (4 × C14:0) + C16:0)/(Σ*n*-3 PUFA + Σ*n*-6 PUFA + Σ MUFA)
TI = (C14:0 + C16:0 + C18:0)/((0.5 × C18:1) + (0.5 × other MUFA) + (0.5 × Σ*n*-6 PUFA) + (3 × Σ*n*-3 PUFA) + Σ*n*-3 PUFA/Σ*n*-6PUFA)

The hypocholesterolemic/hypercholesterolemic ratio (H/H) was calculated according to Ivanova and Hadzhinikolova (2015) [49]:H/H = (C18:1*n*-9 + C18:2*n*-6 + C18:3*n*-3)/(C12:0 + C14:0 + C16:0)(1)

#### 2.2.5. Statistical Analysis

The statistical analyses were performed using the Statistica version 13.1 program (Statsoft, Cracow, Poland) [50]. All statistically significant differences were calculated at *p* < 0.05. Due to the data being defined as having a normal distribution (the Shapiro–Wilk test was performed), to determine the differences between mean values, Duncan’s test was used.

## 3. Results and Discussion

### 3.1. Fatty Acid Composition and Lipid Quality Indices in the Fat from the Analyzed Products

The fatty acid compositions of the fats extracted from the analyzed products are presented in Table 1 and the sum of fatty acids and lipid quality indices are presented in Table 2. In milk fat, saturated fatty acids (SFAs) are the predominant class of fatty acids (FA) [51]. They are unique, as they comprise a wide and complex variety of SFAs, including short-chain fatty acids (SCFAs), odd-chain fatty acids (OCFAs), and branched-chain fatty acids (BCFAs) [3,4]. In the fat from the analyzed products, the content of saturated fatty acids (SFAs) varied (Table 2), being significantly (*p* < 0.05) lower in hard cheeses (on average, 356.44 mg/g of fat ± 40.29) compared to the other analyzed dairy products (Table 2). The highest content of these acids (on average, 502.78 mg/g of fat ± 17.34) was found in the fat extracted from the analyzed butter samples, which had a significantly (*p* < 0.05) higher content of branched-chain fatty acids (BCFAs), odd-chain fatty acids (OCFAs), and short-chain fatty acids (SCFAs) (Table 2). The conducted research showed that in fat extracted from white-mold and blue-veined cheeses, the MUFA content was at a similar level (Table 1). A significantly higher (*p* < 0.05) content of these acids was in the fat from the analyzed butters (on average 225.16 mg/g of fat ± 5.95) and was significantly lower in the fat from the analyzed hard cheeses (on average, 146.21 mg/g of fat ± 23.66) (Table 2). Oleic acid (*cis*9 C18:1) was the main MUFA acid present in the fat extracted from all the analyzed products (Table 1). This acid exhibits anti-cancer, anti-inflammatory, and anti-atherogenic properties [41]. The fat extracted from butter contained significantly higher (*p* < 0.05) amounts of PUFAs (on average, 24.89 mg/g of fat ± 1.61) compared to the other analyzed products (Table 2). The lowest PUFA contents were found in the fat extracted from hard cheeses (on average, 16.07 mg/g of fat ± 2.92). The fat extracted from butters and white-mold cheeses had significantly higher (*p* < 0.05) contents of *n*-3 acids and a lower *n*-6/*n*-3 ratio compared to the fat from hard cheeses and blue-veined cheeses (Table 2). The fat extracted from butter had the highest content of *n*-6 PUFAs (on average, 13.06 mg/g of fat ± 0.89). The content of these acids in the fat extracted from white-mold and blue-veined cheeses was significantly lower (on average, 10.70 ± 1.08 and 10.64 mg/g of fat ± 1.18, respectively). The lowest *n*-6 PUFA contents were determined in the fat extracted from hard cheeses (on average, 9.14 mg/g of fat ± 1.91).

Consuming dairy products that contain high levels of saturated fatty acids is often associated with the development of many diseases. However, given that food is composed of a number of different fatty acids, each of which may affect lipoprotein metabolism in a different way and carry the risk of developing many diseases, it indicates that the relationship between SFA and the occurrence of these diseases may be less obvious than expected [52,53,54]. The short-chain fatty acids (SCFA) present in milk fat are important in promoting human health. Among other things, butyric acid has been shown to be anti-inflammatory and to prevent the progression of colorectal and mammary cancer [55,56]. In turn, branched-chain fatty acids (BCFA) have been demonstrated to elicit anti-carcinogenic effects [57,58]. Research carried out in 2019 showed that the average content of SCFAs in the fat from commercial cheeses purchased from May to June was 62.04 mg/g of fat, and of SFAs, 404.96 mg/g of fat. In the fat extracted from the butter samples, these contents were 100.24 mg/g of fat and 505.78 mg/g of fat, respectively [36]. The average MUFA content in these cheeses was 179.90 mg/g of fat and it reached 222.00 mg/g of fat in the fat extracted from the butter samples. The average content of PUFAs in the fat extracted from these products was at a similar level [36]. Dietary *n*-3 PUFAs can be helpful for heart disease prevention and immune response improvement. Linolenic acid (C18:3) has been shown to exhibit anti-carcinogenic and anti-atherogenic properties [59,60,61], whereas *n*-6 PUFAs are said to improve sensitivity to insulin and thus reduce the incidence of type 2 diabetes [62]. The proportions of the individual groups of fatty acids in food products are of particular importance from a nutritional point of view. Excessive amounts of *n*-6 polyunsaturated fatty acids (PUFA) and a very high *n*-6/*n*-3 ratio in a diet promotes the pathogenesis of many diseases, whereas increased levels of *n*-3 PUFA (a low *n*-6/*n*-3 ratio) exert suppressive effects [47,63,64,65].

In the presented study, the value of the atherogenicity index (AI) determined for hard cheeses was 3.40 ± 0.43 and was significantly higher (*p* < 0.05) compared to the butters (3.09 ± 0.15) and white-mold and blue-veined cheeses (2.69 ± 0.18 and 2.72 ± 0.04, respectively). The thrombogenicity index (TI) value was the highest in hard cheeses, i.e., 4.02 ± 0.42 (Table 2). A significantly (*p* < 0.05) lower value was found in the analyzed mold cheeses and butters. The white-mold and blue-veined cheeses were characterized by similar values of H/H (0.26) (Table 2). A significantly higher H/H ratio was found in the analyzed hard cheeses and butters (0.43 and 0.46, respectively). In the fat extracted from cheeses bought on the Polish market from May to June 2019, the AI was 3.19 and the TI was 3.80. In the fat extracted from butter, the values of these indicators were 3.12 and 3.63, respectively [36]. In the Camembert cheeses analyzed by Adamska et al. [39], the AI ranged from 2.81 to 3.32 and TI ranged from 3.09 to 3.77. In the Brie cheeses, the AI ranged from 2.80 to 3.31 and TI ranged from 3.15 to 3.69. The AI in summer butters analyzed by Blaško et al. [66] was 2.19. In winter butters, it was 2.72. As reported by Ulbrichta and Southgate [47], the AI and TI are better indicators of the atherogenic and thrombogenic potential of a diet than the PUFA/SFA ratio. These indicators take into account the various effects that individual fatty acids can have on human health. The AI and TI indices indicate the likelihood of an increase in the incidence of pathogenic events such as atherosclerosis and/or thrombus formation. The TI value shows the tendency of blood clots to form in the blood vessels. In contrast, the AI value indicates a relationship between the sum of the major saturated fatty acids (SFA) and the sum of the major classes of unsaturated fatty acids (UFA). A low AI value indicates that milk and dairy products could provide protection against coronary heart disease. The higher the AI, the more atherogenic components in the diet.

According to Wang et al. [67], the H/H ratio is related to the functional activity of fatty acids in the metabolism of lipoproteins for the transport of cholesterol in the plasma and to the risk of developing cardiovascular diseases. Higher values of this ratio are desirable. The H/H ratio determined for the fat extracted from hard cheeses and butters bought on the Polish market from May to June 2019 was 0.45 [36].

### 3.2. The Contents of cis9trans11 C18:2 (CLA) and trans C18:1 and C18:2 Fatty Acids in the Fat from the Analyzed Products

The contents of *cis*9*trans*11 C18:2 (CLA) and the *trans* isomers of C18:1 and C18:2 acids in the fat extracted from hard cheeses, blue cheeses, and butter are shown in Figure 1. The acid *cis*9*trans*11 C18:2 (CLA) is the main isomer of CLA in food. In milk and dairy products, it accounts for over 80–90% of the total CLA content. [3,68,69]. The analyzed products had different amounts of this acid. Conjugated linoleic acid *cis*9*trans*11 C18:2 (CLA) has a number of health-promoting properties, including anti-carcinogenic, antiatherosclerotic, antioxidant, and anti-inflammatory effects [70,71,72,73,74,75,76]. Its content in milk fat ranges from 2.3 to 6.0 mg/g of fat [77,78] and may be affected by many factors such as the animal feeding regime, breed, and age, as well as the lactation period [79,80,81,82,83,84]. The content of CLA in dairy products (cheeses and fermented drinks) is influenced by the conditions of technological processes, the additives used, the activity of the starter cultures added, and the ripening time [85,86,87].

The conducted research showed that the fat extracted from white-mold cheeses had the highest mean CLA content (4.12 mg/g of fat ± 1.31) (Figure 1). In the fat from butter, the mean content was 3.63 mg/g of fat ± 0.25, whereas significantly (*p* < 0.05) lower contents were determined in the fat from the blue-veined cheeses and hard cheeses, i.e., 2.89 mg/g ± 0.23 fat and 2.50 mg/g of fat ± 0.52, respectively. According to Seçkin et al. [88], the CLA content ranged from 1.50 mg/g of fat to 3.63 mg/g of fat in Turkish processed cheeses, and from 2.85 mg/g to 4.67 mg/g of fat in butter. The mean content of CLA in the fat extracted from cheeses bought on the Polish market from May to June 2019 was 3.44 mg/g of fat, and in the fat extracted from butter, it was 3.63 mg/g of fat [36]. In the mold-type cheeses studied by Lin et al. [89], the CLA content was 4.75 mg/g of fat in Brie cheese and 4.87 mg/g of fat and 7.96 mg/g of fat in blue cheeses. In white-mold cheeses (seven Camembert-type cheeses and three Brie-type cheeses) tested by Białek and Tokarz [90], the CLA content ranged from 1.74 mg/g of fat to 3.07 mg/g of fat, and in the blue-veined ones, it ranged from 1.66 mg/g of fat to 2.39 mg/g of fat. The average content of CLA in the Gorgonzola-type cheeses analyzed by Prandini et al. [35] was 5.16 mg/g of fat, whereas, in the Brie cheese studied by Nunez and Torres [91], it was 3.8 mg/g of fat.

The average total content of *trans* isomers of C18:1 acid was the highest in the fat extracted from white-mold cheeses (18.15 mg/g of fat). A similar content of these isomers was found in the fat extracted from butter (17.94 mg/g of fat) (Figure 1), while it was significantly (*p* < 0.05) lower in the fat extracted from the blue-veined cheeses and hard cheeses. Monounsaturated fatty acids with 18 carbon atoms are the most important natural *trans* fatty acids (TFA) in the human diet. The major *trans* C18:1 isomer in milk fat, i.e., *trans*11 acid (vaccenic acid, VA), has been reported to exert anti-tumor and antiatherosclerotic effects [92]. The highest content of *trans* C18:2 isomers was found in the fat extracted from butter (4.77 mg/g of fat). The white-mold and blue-veined cheeses had significantly (*p* < 0.05) lower contents of these isomers (Figure 1).

## 4. Conclusions

The presented research provided basic knowledge on the content of the fatty acids and lipid quality indices of dairy products. It demonstrated that the fat extracted from the analyzed cheeses and butters had various contents of fatty acid groups, conjugated linoleic acid *cis*9*trans*11 C18:2, and *trans* C18:1 and C18:2 isomers, as well as various values of the lipid quality indices. This information is important for consumers who plan to follow a healthy and rational diet, as they are increasingly more aware of the impact of consumed products on their health and therefore want to choose products with high sensory values, high quality, and appropriate health-promoting potential.

## Figures and Tables

**Figure 1 animals-12-03424-f001:**
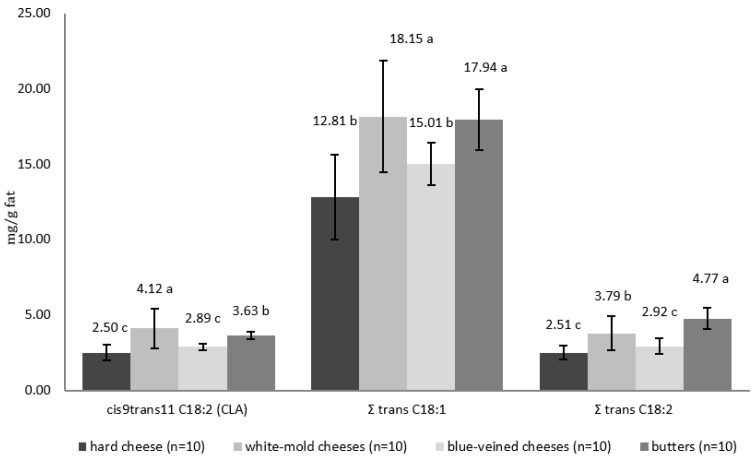
The contents of *cis*9*trans*11 C18:2 (CLA), the sum of *trans* C18:1 and C18:2 in the fat from the analyzed products (mg/g of fat). ^a,b,c^—values with different letters differ significantly (*p* < 0.05).

**Table 1 animals-12-03424-t001:** Fatty acid composition in fat extracted from analyzed products (mg/g of fat).

Fatty Acid	Hard Cheeses			White-Mold Cheeses			Blue-Veined Cheeses			Butters		
n	10			10			10			10		
	Mean	±SD	Min–Max	Mean	±SD	Min–Max	Mean	±SD	Min–Max	Mean	±SD	Min–Max
C4:0	17.55	3.27	14.66–24.56	19.69	2.52	15.92–23.22	17.44	1.61	16.18–20.47	33.09	2.21	30.78–38.67
C6:0	12.72	2.16	10.45–17.35	13.45	1.26	11.74–15.42	12.51	1.23	11.82–14.88	22.72	1.48	20.86–26.26
C8:0	7.75	1.51	5.63–10.86	8.41	0.67	7.69–9.55	7.91	0.82	7.39–9.48	14.09	0.93	12.79–15.95
C10:0	18.82	3.14	12.97–24.75	19.58	1.54	18.22–22.46	18.82	1.91	17.51–22.46	32.04	2.55	31.78–36.33
C10:1	1.89	0.34	1.41–2.58	2.15	0.14	1.95–2.36	2.01	0.18	1.89–2.36	3.20	0.27	2.91–3.74
C12:0	21.81	4.06	15.63–28.58	23.13	1.85	21.75–26.65	22.30	2.25	20.66–26.57	36.09	3.17	31.17–41.60
C12:1	0.49	0.08	0.36–0.63	0.25	0.04	0.19–0.32	0.20	0.02	0.17–0.22	0.30	0.04	0.26–0.35
C13:0 *iso*	0.57	0.09	0.46–0.72	0.51	0.04	0.45–0.56	0.47	0.05	0.44–0.57	0.95	0.10	0.80–1.10
C13:0	0.67	0.10	0.56–0.84	0.62	0.09	0.54–0.79	0.57	0.05	0.53–0.66	2.02	0.21	1.68–2.31
C14:0 *iso*	0.74	0.18	0.51–1.17	0.91	0.13	0.69–1.06	0.77	0.06	0.71–0.86	1.09	0.15	0.77–1.34
C14:0	76.27	8.14	93.87–93.43	79.25	7.11	71.14–92.01	75.08	7.16	70.21–88.78	107.83	6.19	103.30–123.16
C15:0 *iso*	1.46	0.33	1.10–2.20	1.87	0.26	1.49–2.14	1.55	0.12	1.41–1.70	2.24	0.22	1.92–2.55
C15:0 *aiso*	2.96	0.60	2.15–4.04	3.69	0.36	3.10–4.03	3.28	0.24	3.06–3.72	4.52	0.52	3.79–5.52
C14:1	6.31	1.10	4.74–8.28	7.29	0.59	6.52–7.97	7.05	0.64	6.63–8.27	10.06	0.83	9.41–11.07
C15:0	7.06	1.37	5.16–9.64	8.34	0.68	7.49–9.39	7.78	0.76	7.13–9.19	10.96	0.67	10.25–12.32
C16:0 *iso*	1.81	0.42	1.39–2.74	2.04	0.24	1.68–2.24	1.93	0.19	1.80–2.28	2.62	0.26	2.07–3.04
C16:0	200.05	23.24	173.45–244.21	218.09	18.46	197.22–248.26	208.45	20.10	195.39–247.02	277.79	11.35	262.46–299.96
C17:0 *iso*	2.12	0.38	1.57–2.90	2.86	0.42	2.24–3.36	2.32	0.18	2.11–2.53	3.48	0.28	3.07–3.81
C17:0 *aiso*	1.05	0.19	0.81–1.45	1.37	0.24	1.04–1.70	1.22	0.10	1.15–1.41	3.81	0.47	3.19–4.66
C16:1	9.65	2.07	7.30–14.51	13.65	0.96	12.63–15.34	13.34	1.44	12.19–16.04	15.26	1.41	13.41–17.83
C17:0	4.31	0.97	3.67–6.01	0.51	0.23	0.30–0.83	0.35	0.03	0.33–0.40	6.44	0.59	5.61–7.18
C17:1	1.42	0.25	1.08–1.95	1.79	0.14	1.58–1.99	1.67	0.14	1.57–1.93	2.29	0.46	1.61–2.76
C18:0	55.60	6.81	41.73–62.28	70.82	8.00	55.03–76.51	65.84	5.90	62.18–77.19	76.75	5.83	69.47–85.65
C18:1 *trans*6 − *trans*9	2.48	0.52	1.71–3.22	2.98	0.39	2.28–3.49	2.95	0.45	2.47–3.67	3.24	0.55	2.74–3.41
C18:1 *trans*10 + *trans*11	7.09	1.63	5.10–10.16	11.16	2.94	7.73–16.19	8.20	0.59	7.55–9.13	10.20	0.98	8.64–11.89
C18:1 *trans* 12	1.61	0.35	1.08–2.18	1.97	0.27	1.58–2.37	2.01	0.35	1.62–2.56	2.07	0.38	1.53–2.22
C18:1 *cis*9	107.47	16.89	83.38–128.08	142.43	15.22	115.88–162.42	134.75	13.71	124.12–160.57	159.71	5.58	153.15–170.82
C18:1 *cis*11	3.54	0.72	2.48–4.60	5.23	0.57	4.35–6.02	5.54	0.79	4.68–6.80	6.85	0.52	6.15-7.64
C18:1 *cis*12	1.51	0.41	0.90–2.23	1.75	0.29	1.46–2.21	1.89	0.36	1.48–2.43	2.64	0.37	2.11-3.15
C18:1 *cis*13	0.48	0.14	0.60–0.71	0.72	0.15	0.49–0.90	0.73	0.11	0.61–0.90	0.88	0.09	0.72-1.03
C18:1 *trans*16	1.63	0.39	1.11–2.29	2.04	0.36	1.56–2.52	1.84	0.26	1.56–2.27	2.43	0.29	1.88-2.88
C19:0	0.91	0.22	0.63–1.28	1.21	0.24	0.86–1.49	1.07	0.16	0.92–1.35	1.09	0.33	0.68-1.39
C18:2 *cis*9*trans*13	1.09	0.26	0.77–1.66	1.31	0.33	0.95–1.86	1.15	0.18	0.96–1.45	1.57	0.36	0.78-2.12
C18:2 *cis*9*trans*12	0.88	0.17	0.66–1.15	1.09	0.18	0.85–1.27	1.01	0.22	0.76–1.34	2.25	0.23	1.96-2.36
C18:2 *trans*11*cis*15	0.54	0.24	0.33–1.04	1.28	0.58	0.75–2.40	0.65	0.03	0.63–0.72	0.95	0.23	0.85-1.18
C18:2 *cis*9*cis*12	9.14	1.91	6.37–10.55	10.70	1.08	9.03–12.07	10.64	1.18	9.52–12.74	13.06	0.89	11.77-14.75
C20:0	0.87	0.17	0.68–1.22	1.05	0.16	0.82–1.28	0.96	0.10	0.89–1.16	1.21	0.18	0.89-1.42
C20:1	0.64	0.12	0.48–0.87	0.82	0.08	0.69–0.93	0.71	0.06	0.68–0.83	0.91	0.14	0.71-1.11
C18:3 *cis*9*cis*12*cis*15	1.92	0.57	1.23–2.88	2.95	0.69	2.38–4.14	1.77	0.27	1.48–2.09	3.41	0.73	2.62-4.38
C18:2 *cis*9*trans*11 (CLA)	2.50	0.52	1.94–3.56	4.12	1.31	3.61–6.61	2.89	0.23	2.60–3.15	3.63	0.25	3.23-4.10

n—number of samples; Mean—mean value; SD—standard deviation; Min—minimum value, Max—maximum value.

**Table 2 animals-12-03424-t002:** Fatty acids content (mg/g of fat) and lipid quality indices in the fat from the analyzed products (Mean ± SD, range).

Fatty Acids		Hard Cheeses	White-Mold Cheeses	Blue-Veined Cheeses	Butters
n		10	10	10	10
Σ SCFA ^1^	Mean	56.84 ^b^	61.12 ^b^	56.68 ^b^	101.93 ^a^
	SD	9.5	5.68	5.51	6.61
	Min–Max	46.18–78.00	53.97–70.65	53.58–67.29	93.12–116.90
Σ BCFA ^2^	Mean	10.70 ^c^	13.24 ^b^	11.55 ^b,c^	18.70 ^a^
	SD	2.13	1.51	0.84	1.84
	Min–Max	8.00–15.20	10.77–15.10	10.78–13.07	15.88–22.00
Σ OCFA ^3^	Mean	12.95 ^b^	11.03 ^c^	9.80 ^c^	20.51 ^a^
	SD	2.62	1.17	1.03	1.41
	Min–Max	9.43–17.47	9.68–12.83	8.88–11.68	18.67–21.41
Σ SFA ^4^	Mean	356.44 ^c^	416.62 ^b^	394.31 ^b^	502.78 ^a^
	SD	40.29	34.56	37.3	17.34
	Min–Max	307.23–430.54	366.88–471.64	370.02–465.87	475.39–524.61
Σ MUFA ^5^	Mean	146.21 ^c^	194.24 ^b^	182.88 ^b^	225.16 ^a^
	SD	23.66	20.32	18.71	5.95
	Min–Max	101.43–175.77	159.56–221.47	168.08–217.97	216.81–237.00
Σ PUFA ^6^	Mean	16.07 ^c^	21.57 ^b^	18.22 ^c^	24.89 ^a^
	SD	2.92	3.38	1.65	1.61
	Min–Max	12.50–21.02	17.13–26.56	16.99–21.34	21.90–28.06
*n*-3	Mean	1.92 ^b^	2.95 ^a^	1.77 ^b^	3.41 ^a^
	SD	0.57	0.69	0.27	0.73
	Min–Max	1.73–2.30	2.38–4.14	1.48–2.09	2.70–4.50
*n*-6	Mean	9.14 ^c^	10.70 ^b^	10.64 ^b^	13.06 ^a^
	SD	1.91	1.08	1.18	0.89
	Min–Max	7.82–13.01	9.03–12.07	9.52–12.74	11.80–15.05
*n*-6/*n*-3	Mean	5.05 ^a,b^	3.76 ^b^	6.17 ^a^	4.01 ^b^
	SD	1.52	0.74	1.31	0.93
	Min–Max	3.60–8.00	2.42–4.47	4.56–7.41	2.83–5.00
AI ^7^	Mean	3.40 ^a^	2.69 ^c^	2.72 ^c^	3.09 ^b^
	SD	0.43	0.17	0.04	0.15
	Min–Max	3.01–4.01	2.44–2.95	2.69–2.77	2.86–3.32
TI ^8^	Mean	4.02 ^a^	3.31 ^b^	3.42 ^b^	3.57 ^b^
	SD	0.42	0.2	0.02	0.12
	Min–Max	3.65–4.54	2.94–3.53	3.39–3.44	3.40–3.80
HH ^9^	Mean	0.43 ^a^	0.26 ^b^	0.26 ^b^	0.46 ^a^
	SD	0.05	0.02	0.01	0.02
	Min–Max	0.37–0.50	0.23–0.29	0.24–0.27	0.42–0.51

n—number of samples; Mean—mean value; SD—standard deviation; Min—minimum value; Max—maximum value; ^a,b,c^—values denoted in rows by different letters differ significantly (*p* < 0.05); ^1^ Σ SCFA—sum of short-chain fatty acids (C4:0–C10:0); ^2^ Σ BCFA—sum of branched-chain fatty acids; ^3^ Σ OCFA—sum of odd-chain fatty acids; ^4^ Σ SFA—sum of saturated fatty acids (BCFA, OCFA, MCFA, LCFA without SCFA); ^5^ Σ MUFA—sum of monounsaturated fatty acids; ^6^ Σ PUFA—sum of polyunsaturated fatty acids; ^7^ AI (Index of Atherogenicity); ^8^ TI (Index of Thrombogenicity); ^9^ H/H (hypocholesterolemic/hypercholesterolemic ratio).

## Data Availability

Data is contained within the article.

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
