# Peer review of "Cheese and Butter as a Source of Health-Promoting Fatty Acids in the Human Diet"

_animals, 2022, doi:10.3390/ani12233424_

Round 1
Reviewer 1 Report
General note:
The work requires thorough improvement.
Please carefully analyze the work and complete the references. Many fragments of the work contain information that should be referenced. It concerns, among others, verses: 57-58; 58-60; 60-62; 73.
The work should be structured so that the same fragments or descriptions of the same ingredients in the analyzed products are not repeated several times in the text.
Please clearly state the purpose of the study because the content of fatty acids in cheeses is known and available in many studies, including Polish tables of the composition of the nutritional value of food products.
Detailed comments:
I propose to edit the Introduction section so that it corresponds to the adopted purpose of the work. The work is chaotic, and after a few sentences, it loses the context of the work, followed by repetitions and re-references to the goal of the work.
Please eliminate repetitions, eg lines 57-58; 83-85; 103-104 and 113-114. The selected passages are practically the same.
Lines 121-123. The author states: "Owing to a high intake of butter and cheeses by Polish consumers and a rich assortment of these products available on the Polish market, the assessment of their quality is of great importance". Has there been no research in this area so far? Please indicate which ones and link them to the author's research.
Materials and Methods
Line 142: Please explain: "bought at the same time".
It should be noted in this section that standard deviation is included in the calculations.
Results and discussion.
This passage is very chaotic. First, the author should present his research results and then discuss them with the literature data. Meanwhile, the author cites data from other authors, referring them to the percentage data and not to the amount given in mg / g.
If it is not known precisely what the purpose of the work is, it isn't easy to refer to the research results and discussions.
Author Response
Response to Reviewer 1 Comments
Thank you very much for the review of the manuscript. Below are responses to the Reviewer's comments. All changes have been included in the manuscript.
Please carefully analyze the work and complete the references. Many fragments of the work contain information that should be referenced. It concerns, among others, verses: 57-58; 58-60; 60-62; 73.
As suggested by the Reviewer, the work was checked and the bibliography supplemented:
References have been added in line 61, 63, 66, 67.
Please clearly state the purpose of the study because the content of fatty acids in cheeses is known and available in many studies, including Polish tables of the composition of the nutritional value of food products.
According to the Reviewer's suggestion, the purpose of the work has been improved. In the work added:
L139-144: Therefore given that the fatty acid profile of milk and dairy products is an important factor affecting their nutritional value, the aim of this study was to determine the composition of fatty acids, with special emphasis on the concentrations of conjugated linoleic acid cis9trans11 C18:2 (CLA), trans C18:1 and C18:2 isomers, and the lipid quality indices in selected cheeses (hard and mold) and butters commonly available to consumers in retail sale on the Polish market.”
Detailed comments:
I propose to edit the Introduction section so that it corresponds to the adopted purpose of the work. The work is chaotic, and after a few sentences, it loses the context of the work, followed by repetitions and re-references to the goal of the work.
According to the Reviewer's suggestion, the introduction of the paper has been edited.
Please eliminate repetitions, eg lines 57-58; 83-85; 103-104 and 113-114. The selected passages are practically the same.
As suggested by the Reviewer, the work has been checked, all repetitions in the text have been removed: L87-88, L107-108.
Lines 121-123. The author states: "Owing to a high intake of butter and cheeses by Polish consumers and a rich assortment of these products available on the Polish market, the assessment of their quality is of great importance". Has there been no research in this area so far? Please indicate which ones and link them to the author's research.
As suggested by the Reviewer, this part of the work has been corrected . The following text was added:
L125-133: ”The composition of fatty acids in dairy products, including hard cheeses made from the milk of various ruminants was a subject of many studies [29-38], whereas other reports focused on the assessment of the fatty acid profile in blue cheeses [36,38,39,40] and butters [36, 41-44]. A rich assortment of butter and cheese available on the Polish market is largely attributed to a high intake and popularity of those products in Polish consumers. Taking into account the fact that the composition of these products may change as a result of changes in the quality of the raw material or various parameters used in technological processes, constant monitoring of their quality is of great importance from a nutritional point of view.
Materials and Methods
Line 142: Please explain: "bought at the same time".
As suggested by the Reviewer, this has been changed to: “All of the analyzed products were made from cow’s milk, produced by various manufacturers in Poland and were available on the Polish market from June to July.”
Results and discussion.
This passage is very chaotic. First, the author should present his research results and then discuss them with the literature data. Meanwhile, the author cites data from other authors, referring them to the percentage data and not to the amount given in mg / g.
According to the Reviewer's suggestion, this fragment of the work has been edited. First, the results obtained in the study were presented, and then they were compared with literature data. As suggested, inappropriate comparisons have been removed.

Reviewer 2 Report
Mostly and detailed comments:
The "Simlpe summary" must be shortened as required by Animals Journal (to consist of no more than 200 words).
The "Abstract" also should be shortened (total of about 200 words maximum - guide for authors). I would suggest removing lines 49-53, because it was not the subject of the study. This information can be found in the chapter „Result and Discussion” or „Conlusion”.
The „Keywords” basically repeat the words in the title, without expanding the information to other elements of the work. Please reword.
In the chapter “Introduction” (line 63) reduce the number of items cited to a maximum of 4, as the impact of nutrition has not been studied.
“Statistical analysys” – there is no information on whether all data were tested for normality. In case the distribution would not be normal, another test had to be used. Lines 184-185 – Cracow insted of Kraków.
Table 1 - no statistical differences between the groups are included (about the fact that it was noted indicated Fig. 1 – c9t11C18:2). In column one I would suggest position fatty acids by increasing number of carbon atoms. Minor editorial corrections require lines C4:0 and cis9trans11C18:2 (CLA).
Paragraph „3.2. Fatty acid composition and lipid quality indices in the fat from the analyzed products” is required a more substantive discussion. The cited items mostly deal with the beneficial or negative effects of fatty acid groups on the human body, while missing confronting results regarding the fatty acid profile of cheese and butter from other authors' studies.
Throughout the work, remove the word splitting function.
Citing more than two items: [x–y] instead of [x-y], for example line 63.
The title of the paper and subsections according to the requirements of Animals Journal (letter size).
Lines 447 and 494-495: Remove underlines and hyperlinks.
Improvements, as required by Animals Journal, need some bibliographic descriptions, for example 5, 10, etc.
Author Response
Response to Reviewer 2 Comments
Thank you very much for the review of the manuscript. Below are responses to the Reviewer's comments. All changes have been included in the manuscript.
Mostly and detailed comments:
The "Simlpe summary" must be shortened as required by Animals Journal (to consist of no more than 200 words).
As suggested by the Reviewer, the "Simple summary" has been shortened.
The "Abstract" also should be shortened (total of about 200 words maximum - guide for authors). I would suggest removing lines 49-53, because it was not the subject of the study. This information can be found in the chapter „Result and Discussion” or „Conlusion”.
As suggested by the Reviewer, the "Abstract" has been shortened.
The „Keywords” basically repeat the words in the title, without expanding the information to other elements of the work. Please reword.
As suggested by the Reviewer, the ,,Title” and " Keywords " have been changed. L: 55-56
In the chapter “Introduction” (line 63) reduce the number of items cited to a maximum of 4, as the impact of nutrition has not been studied.
As suggested by the Reviewer, the number of references has been limited.
“Statistical analysys” – there is no information on whether all data were tested for normality. In case the distribution would not be normal, another test had to be used. Lines 184-185 – Cracow insted of Kraków.
As suggested by the Reviewer, the “Statistical analysis” part of the paper it has been supplemented:
L: 203-206 “All statistically significant differences were calculated at P < 0.05. Due to the data being defined as having a normal distribution (Shapiro-Wilk test was performed), to determine the differences between mean values the Duncan’s test was used”
Table 1 - no statistical differences between the groups are included (about the fact that it was noted indicated Fig. 1 – c9t11C18:2). In column one I would suggest position fatty acids by increasing number of carbon atoms. Minor editorial corrections require lines C4:0 and cis9trans11C18:2 (CLA).
Statistical differences are given only in Table 2 and Figure 1. The data in Table 1 is for information only. As suggested by the Reviewer, in Table 1 the positioning of fatty acids in the first column has been changed according to the increasing number of carbon atoms. Minor corrections in given lines have been made.
Paragraph „3.2. Fatty acid composition and lipid quality indices in the fat from the analyzed products” is required a more substantive discussion. The cited items mostly deal with the beneficial or negative effects of fatty acid groups on the human body, while missing confronting results regarding the fatty acid profile of cheese and butter from other authors’ studies.
The composition of fatty acids in dairy products, including hard cheeses, blue cheeses and butters, was assessed by many authors. However, in most of these studies, the acid composition was reported as % of the total fatty acid composition and not as mg/g of fat. In the paper includes the following comparisons:
,,Own research carried out in 2019 showed that the average content of SCFAs in the fat from commercial cheeses purchased from May to June was 62.04 mg/g fat, and of SFAs 404.96 mg/g fat. In the fat extracted from the butter samples, these contents were 100.24 mg/g fat and 505.78 mg/g fat, respectively [36]. The average MUFAs content in these cheeses was 179.90 mg/g fat and it reached 222.00 mg/g fat in the fat extracted from the butter samples. The average content of PUFAs in the fat extracted from these products was at a similar level [36].”
Throughout the work, remove the word splitting function.
Citing more than two items: [x–y] instead of [x-y], for example line 63.
Lines 447 and 494-495: Remove underlines and hyperlinks.
Improvements, as required by Animals Journal, need some bibliographic descriptions, for example 5, 10, etc.
Thanks for noticing these errors, this has been corrected.

Round 2
Reviewer 1 Report
I want to thank the author for the changes made in accordance with the reviewer's suggestion.
The work has been modified and significantly improved.
The reviewer asks the author for a thorough check of the work because there are unclear fragments due to the removal of text fragments (introduced changes), e.g. lines 272-276.